# Data as a Lever: A Neighbouring Datasets Perspective on Predictive Multiplicity

## Abstract

Multiplicity—the existence of distinct models with comparable performance—
has received growing attention in recent years. While prior work has largely
emphasized modelling choices, the critical role of data in shaping multiplicity
has been comparatively overlooked. In this work, we introduce a neighbouring
datasets framework to examine the most granular case: the impact of a single-data-
point difference on multiplicity. Our analysis yields a seemingly counterintuitive
finding: neighbouring datasets with greater inter-class distribution overlap exhibit
lower multiplicity. This reversal of conventional expectations arises from a shared
Rashomon parameter, and we substantiate it with rigorous proofs.

Building on this foundation, we extend our framework to two practical domains:
active learning and data imputation. For each, we establish natural extensions
of the neighbouring datasets perspective, conduct the first systematic study of
multiplicity in existing algorithms, and finally, propose novel multiplicity-aware
methods, namely, *multiplicity-aware data acquisition* strategies for active learning
and *multiplicity-aware data imputation* techniques.

## 1 Introduction

Predictive multiplicity refers to the phenomenon of a set of "good models" (the Rashomon set),
typically defined as models whose performance exceeds a given threshold (the Rashomon parameter),
learning distinct decision boundaries and therefore producing conflicting predictions for the same
individual (Marx et al., 2020; Black et al., 2022; Breiman, 2001; Ganesh et al., 2025).

Multiplicity has been a point of concern for many, as decisions that affect individuals lack adequate
justification when a model is chosen arbitrarily from the Rashomon set (Black et al., 2022; Gomez
et al., 2024; Watson-Daniels et al., 2024; Sokol et al., 2024). At the same time, multiplicity is also
championed as a counterbalance to monoculture, where reliance on a single dominant system can
systematically deny individuals access to critical resources, and multiplicity can introduce much
needed diversity (Creel & Hellman, 2022; Jain et al., 2024b;a; Kleinberg & Raghavan, 2021).

Recent work by Gur-Arieh & Lee (2025) attempts to bring together these two strands of research
by identifying distinct settings in which one might prefer consistency versus arbitrariness. When
dealing with applications that are normative or have multiple actors, higher multiplicity introduces
diversity, and developer choices that might block this diversity can result in blanket rejections for
individuals (Creel & Hellman, 2022). On the other hand, when working with applications that are
factual or have only a single actor, consistency, even though built on an arbitrary choice, is preferred
to maintain trust in these systems (Gur-Arieh & Lee (2025) call it an 'arbitrarily consistent' choice).

Irrespective of the direction, controlling multiplicity requires understanding how developer choices
shape downstream outcomes (Ganesh et al., 2025). While existing work has primarily examined how
choices during model training influence predictive multiplicity (Black et al., 2022), the role of data
processing remains largely overlooked. This gap may stem from the difficulty of mapping how data
processing decisions affect downstream models without actually training them (Koh et al., 2019), or
from the prevailing norm in the literature of relying on pre-cleaned and already processed datasets
rather than interrogating the cleaning choices themselves (Paullada et al., 2021).

Consider, for example, a task with missing values for predicting an individual's income (Ding et al.,
2021). Using our multiplicity-aware imputation methods (more details in §6), we find that the choice

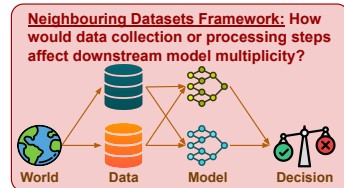

Figure 1: Our neighbouring datasets framework alongside model and dataset multiplicity frameworks.

of imputation can shift downstream multiplicity from $14\%$ to $24\%$, i.e., up to $10\%$ of the dataset is affected by this one data processing choice. Income predictors are used in applications such as loan approval or hiring, where multiplicity can be helpful to prevent monoculture (Gur-Arieh & Lee, 2025). Thus, a poor imputation choice can potentially result in a blanket rejection for up to $10\%$ of the data, not recoverable irrespective of choices made during model training. Clearly, choices made during data processing play a significant role in downstream multiplicity.

While recent frameworks like dataset multiplicity (Meyer et al., 2023) study noise in the data while keeping the training pipeline fixed, we argue that isolating either model or dataset multiplicity will give us an incomplete picture. In our work, we instead focus on how different data processing choices—creating *neighbouring datasets*—affect model multiplicity (see Figure 1 for an illustration).

The perspective of neighbouring datasets, inspired by the literature in differential privacy (Dwork, 2006), pops up repeatedly and naturally in many data processing scenarios, such as data acquisition for active learning (Ren et al., 2021; Aggarwal et al., 2014), data imputation (Miao et al., 2022), and handling outliers (Aguinis et al., 2013), among others. Data processing rarely transforms a dataset entirely; instead, it introduces incremental changes that can still have significant downstream effects. For instance, consider data imputation, where different techniques may fill the missing values in distinct ways. However, the majority of the data is not missing and thus remains unchanged. Hence, data imputation can be seen as a choice between various *neighbouring datasets*.

**Contributions.** By framing our study through the lens of neighbouring datasets, we provide a unified framework that accommodates many frequently studied problems in data processing, allowing us to systematically examine developers' choices and their influence on multiplicity. We then apply the insights derived from this perspective to two well-established subdomains of data processing, active learning and data imputation, highlighting the trends of downstream multiplicity as well as designing new algorithms offering control over multiplicity. More specifically, our main contributions are,

1. **Neighbouring Datasets Framework:** A novel unified framework to study the impact of various data processing choices on multiplicity (§3). We formalize neighbouring datasets for deeper theoretical insights in controlled settings and practical extensions in real-world applications.

2. **Reversed Multiplicity Trends under a Shared Rashomon Parameter:** Theoretical insights into neighbouring datasets and multiplicity reveal a surprising result: under a shared Rashomon parameter, less separability leads to lower multiplicity (§4). This reverses expected trends based on prior work (Watson-Daniels et al., 2023b; Semenova et al., 2024). Without contradicting existing literature, this reversal occurs due to the use of a shared Rashomon parameter across neighbouring datasets, highlighting how these frameworks do not capture multiplicity trends in data processing.

3. **Multiplicity and Active Learning:** We investigate active learning from the lens of neighbouring datasets, performing the first empirical study of multiplicity in active learning, as well as using our theoretical insights to propose new multiplicity-aware active learning algorithms (§5). Our experiments reveal consistent trends of less separability leading to lower multiplicity, even beyond the assumptions of our theoretical analysis, further strengthening the value of our framework.

4. **Multiplicity and Data Imputation:** We repeat our study for another important data processing task, data imputation, and observe a similar set of contributions and trends as in active learning (§6). Interestingly, we also find that more missing data amplifies the influence of the imputation on multiplicity, thus, in turn, giving stronger control to our multiplicity-aware algorithms.

Figure 2: Examples of neighbouring datasets in several data preparation and processing pipelines.

## 2 RELATED WORK

**Multiplicity and Rashomon Sets.** The literature on multiplicity has grown rapidly (Ganesh et al., 2025), with a particular focus on predictive multiplicity (Marx et al., 2020; Cooper et al., 2022; Watson-Daniels et al., 2024). Through extension to new forms of multiplicity (Watson-Daniels et al., 2023a; 2024; Hsu et al., 2024b), development of better tools for auditing and quantifying multiplicity (Hsu et al., 2024b; Kissel & Mentch, 2024; Zhong et al., 2024; Xin et al., 2022; Hsu et al., 2024a; Ganesh, 2024), and deeper investigations into the benefits and harms of multiplicity (Black et al., 2022; Rudin et al., 2024; Gur-Arieh & Lee, 2025), it is evident that multiplicity has become a valuable lens for understanding the ambiguity inherent in learning pipelines.

Yet, despite growing interest, most research continues to concentrate only on modeling decisions during learning (Ganesh et al., 2025), evident also through significant focus on stability analysis of learning algorithms in the literature (Picard, 2021; Lei & Ying, 2020; Bertsimas et al., 2022). In contrast, our work joins a smaller but emerging thread of research that aims to uncover the inherent multiplicity in the datasets themselves (Meyer et al., 2023; Cavus & Biecek, 2024; Semenova et al., 2024; Watson-Daniels et al., 2023b).

**Data and Multiplicity.** Meyer et al. (2023) proposed a framework for dataset multiplicity, showing how noisy data can introduce multiplicity. However, while their focus lies in aggregating variance across datasets using a fixed learning pipeline, we instead investigate and minutely compare variations across datasets and their relationship with downstream multiplicity under changing learning pipelines.

The works closely related to our theoretical analysis are those of Semenova et al. (2024); Watson-Daniels et al. (2023b). Semenova et al. (2024) demonstrate that noisier tasks, i.e., tasks with higher inter-class distribution overlap, exhibit higher multiplicity. Watson-Daniels et al. (2023b) provide similar insights on the low separability of a task as a potential cause of multiplicity. Interestingly, our examination of neighbouring datasets under a shared Rashomon parameter reverses these trends (Semenova et al., 2024; Watson-Daniels et al., 2023b). This is because existing frameworks are designed to compare distinct tasks, and not neighbouring datasets within a single task. Our framework addresses this gap, enabling the study of how data processing affects multiplicity.

On the empirical side, the study by Cavus & Biecek (2024) is most related to our work. They conduct a large-scale empirical analysis of data balancing methods and their effect on multiplicity. We provide a similar analysis for data acquisition and imputation techniques. Furthermore, drawing on our theoretical insights, we also introduce *multiplicity-aware* data processing, which can achieve the lowest (or highest) multiplicity while preserving accuracy.

**Active Learning and Data Imputation.** In this work, we study two components of data processing from the lens of neighbouring datasets, namely active learning and data imputation. Active learning focuses on selecting the data points to label (Ren et al., 2021; Aggarwal et al., 2014), recognizing that labeling is often expensive. On the other hand, data imputation deals with the issue of missing data (Miao et al., 2022). Together, they represent decisions that developers must navigate during data collection and preparation. Although both fields have rich histories of research, to the best of our knowledge, we are the first to study their impact on multiplicity.

## 3 NEIGHBOURING DATASETS

When preparing data, developers routinely make decisions that involve choosing between neighbouring datasets. Examples include: active learning (Ren et al., 2021), where the new data points to label are chosen while the rest of the dataset remains unchanged; data imputation (Miao et al., 2022),

where a few missing values are filled leading to datasets varied in only those data points; and handling outliers (Neale, 2016), where normalizing only affects outliers (see Figure 2 for illustrations).

Making these choices with an awareness of multiplicity allows developers to understand and control the downstream trends. Thus, studying multiplicity for neighbouring datasets can enable multiplicity-aware data collection and preparation practices from the outset and lead to informed decision-making.

## 3.1 PRELIMINARIES: RASHOMON SET AND MULTIPLICITY

Consider a supervised learning setup, with data distribution $\mathbb{D} \equiv \mathbb{X} \times \mathbb{Y}$, where $\mathbb{X}$ represents the feature distribution and $\mathbb{Y}$ represents the label distribution. We sample two datasets independently from the distribution $\mathbb{D}$, the train dataset $D_{train} \equiv (X_{train}, Y_{train}) \sim \mathbb{D}$ and the test dataset $D_{test} \equiv (X_{test}, Y_{test}) \sim \mathbb{D}$. Given a loss function $L(\theta, D)$ for the parameter vector $\theta$ on the dataset $D$, and the Rashomon parameter $\epsilon$, the Rashomon set is defined as (Hsu & Calmon, 2022):

**Definition 3.1** (Rashomon Set). The set of all parameter vectors $\Theta \equiv \{\theta_1, \theta_2, ...\}$, such that the loss defined by $L(\theta_i, D_{train})$ for each parameter vector in the set is less than a given threshold $\epsilon$, i.e.,

$$\Theta_{(D_{train}, \epsilon)} \equiv \{\theta_i \mid L(\theta_i, D_{train}) \leq \epsilon\} \tag{1}$$

The Rashomon set is the set of models that achieve similar loss on the training dataset. We will omit the subscript and refer to the Rashomon set as simply $\Theta$ for brevity. We can then quantify multiplicity as $M(\Theta, D_{test})$, where $M()$ is a multiplicity metric that maps the Rashomon set and the test dataset to a score between $0$ and $1$, representing the severity of prediction conflicts. For instance, we can quantify predictive multiplicity for classification by defining ambiguity (Marx et al., 2020) $M^A()$ as:

**Definition 3.2** (Ambiguity). The ambiguity of a prediction problem over the Rashomon set $\Theta$ is the proportion of points in the test dataset $D_{test}$ that can be assigned a conflicting prediction between two classifiers in the Rashomon set, i.e., $\theta_i, \theta_j \in \Theta$:

$$M^A(\Theta, D_{test}) = \frac{1}{|D_{test}|} \sum_{x \in D_{test}} \max_{\theta_i, \theta_j \in \Theta} \mathbb{1}[f_{\theta_i}(x) \neq f_{\theta_j}(x)] \tag{2}$$

where $f_\theta$ represent the prediction function corresponding to the parameter vector $\theta$. We will denote multiplicity as $M_\Theta$ (for example, ambiguity as $M_\Theta^A$) for brevity. We make a distinction between the Rashomon set created on the train dataset $D_{train}$ and the multiplicity measured on the test dataset $D_{test}$. This is different from the tradition of measuring multiplicity on the train dataset itself (Marx et al., 2020). We argue that this distinction is important in practice, as the phenomenon of several models achieving similar loss and thus forcing an arbitrary choice by the developer occurs during training, while its impact and hence the multiplicity is felt when the model is deployed.

## 3.2 $k$-NEIGHBOURING DATASETS

**Definition 3.3** ($k$-Neighbouring Datasets). Two datasets $D^1, D^2$ of same size, i.e., $|D^1| = |D^2| = n$ are considered $k$-neighbouring if they differ in exactly $k$ data points, i.e.,

$$|D^1| = |D^2| = n \quad \text{and} \quad \left|\{i : D_i^1 \neq D_i^2\}\right| = k \ll n \tag{3}$$

Here, the size of a dataset $|D|$ represents the number of data points present in the dataset.

**Objective:** As previously discussed, the formulation of $k$-neighbouring datasets extends naturally to various data preparation decisions, where the developer has to choose between several neighbouring datasets. The objective, thus, is to facilitate a multiplicity-aware choice in such scenarios. More formally, given two $k$-neighbouring datasets $D_{train}^1, D_{train}^2$, and the Rashomon sets on these datasets denoted by $\Theta^1 \equiv \Theta_{(D_{train}^1, \epsilon)}, \Theta^2 \equiv \Theta_{(D_{train}^2, \epsilon)}$, we aim to compare the multiplicity due to these datasets on a common test set $D_{test}$, i.e., compare the values $M_{\Theta^1}$ and $M_{\Theta^2}$.

Note that we aim to compare and anticipate multiplicity without explicitly constructing Rashomon sets along the way. Each data-processing step can be viewed as selecting among many neighbouring datasets, and generating a full Rashomon set for each of these neighbours would be computationally prohibitive. For instance, creating Rashomon sets separately for each possible imputation of a dataset with missing values. Consequently, explicitly creating these intermediate Rashomon sets is impractical. Our objective, therefore, is to understand how data influences multiplicity while bypassing the explicit construction of Rashomon sets.

# 4 HIGHER OVERLAP LEADS TO A SMALLER RASHOMON SET

Data-driven learning methods typically rely on implicitly approximating the underlying distribution. As a result, learning a classifier is tightly coupled with learning the empirical distribution. Intuitively, when the distributions of various classes in a dataset exhibit greater overlap than those of its neighbouring datasets, the decision boundary becomes more ambiguous and can lead to higher error rates. With a fixed Rashomon parameter $\epsilon$, under appropriate assumptions, such a shift can exclude some models from the Rashomon set, thereby reducing multiplicity under any metric that is monotonic within the Rashomon set (Ganesh et al., 2025). *Monotonicity within the Rashomon set for a metric is defined as multiplicity decreasing or staying the same if models are removed from the Rashomon set. Ambiguity* (Definition 3.2) *is monotonic within the Rashomon set (Ganesh et al., 2025).*

Note, it is vital to emphasize that the insights presented in our work are based on comparisons between neighbouring datasets. This framing is important because it allows us to apply a shared fixed threshold $\epsilon$ across datasets. At first glance, our claim may seem counterintuitive, as higher overlap and higher error rates are typically associated with higher multiplicity (Semenova et al., 2024; Watson-Daniels et al., 2023b). However, this is because when comparing different tasks, the Rashomon sets are defined using a task-dependent threshold $\epsilon$, hence leading to the trends seen in the literature. In contrast, our analysis focuses on neighbouring datasets for the same task, where we argue that the threshold for what constitutes a "good model" should not vary due to data processing choices. In other words, the threshold for a good model remains anchored to the task itself [1]. As we will demonstrate, under this constraint, *higher overlap leads to a smaller Rashomon set*.

## 4.1 THEORETICAL INSIGHTS FOR BINARY CLASSIFICATION

Consider two 1-neighbouring training datasets $D_{train}^1, D_{train}^2$. The learning task is binary classification, i.e., $Y_{train}^1, Y_{train}^2 \in \{0,1\}^n$. To make the class structure explicit, we decompose each training dataset $D_{train}^i \, \forall i \in \{1,2\}$ into its two class-specific subsets $D_{train}^i \equiv 0_{train}^i \cup 1_{train}^i$, where:

$$0_{train}^i \equiv \{(x_j^i, y_j^i) \in D_{train}^i \mid y_j^i = 0\}, \qquad 1_{train}^i \equiv \{(x_j^i, y_j^i) \in D_{train}^i \mid y_j^i = 1\}.$$

For each class-specific subset, we assume the existence of an underlying class-conditional probability distribution, $\mathbb{P}_c^i$ for $c \in \{0,1\}$, i.e., $0_{train}^i \sim \mathbb{P}_0^i, 1_{train}^i \sim \mathbb{P}_1^i$. Using these probability distributions, the overlap between the two classes is measured using the overlapping coefficient defined as:

**Definition 4.1** (Overlapping Coefficient (Inman & Bradley Jr, 1989)). The overlapping coefficient (OVL) between two probability (density) distributions $\mathbb{P}_0^i, \mathbb{P}_1^i$ is defined as:

$$OVL(\mathbb{P}_0^i, \mathbb{P}_0^i) = \int_x min(\mathbb{P}_0^i(x), \mathbb{P}_0^i(x)) \, dx \tag{4}$$

The overlapping coefficient is the complement to total variation distance (TVD) (Dudley, 2018), i.e., $OVL + TVD = 1$. We use $OVL_{train}^i = OVL(\mathbb{P}_0^i, \mathbb{P}_1^i)$ to represent the overlapping coefficient between the two classes in the training dataset.

Under the assumptions of a 0-1 loss function, we show that:

**Theorem 4.1.** Given two 1-neighbouring binary classification datasets $D_{train}^1, D_{train}^2$ which, without loss of generality, differ only at the index 0, i.e., $(x_0^1, y_0^1) \neq (x_0^2, y_0^2)$ and $(x_j^1, y_j^1) = (x_j^2, y_j^2) \, \forall j \neq 0$, and adhere to the following assumptions:

1. Loss of all models in the Rashomon set is higher on one differing data point over another, i.e.,

$$L(\theta, (x_0^1, y_0^1)) \geq L(\theta, (x_0^2, y_0^2)) \quad \forall \theta \in \Theta_{(D_{train}^1, \epsilon)} \cup \Theta_{(D_{train}^2, \epsilon)} \tag{5}$$

2. Loss of the Bayes optimal models $\theta_1^*, \theta_2^*$ follow the same trend as the Rashomon set, i.e.,

$$L(\theta_1^*, (x_0^1, y_0^1)) \geq L(\theta_2^*, (x_0^2, y_0^2)) \tag{6}$$

then we can say that the overlapping coefficient between the two classes will be higher for this dataset, i.e., $OVL_{train}^1 \geq OVL_{train}^2$, and the resulting Rashomon set for this dataset under a common threshold $\epsilon$ will be a subset of the Rashomon set for the other dataset, i.e., $\Theta_{(D_{train}^1, \epsilon)} \subseteq \Theta_{(D_{train}^2, \epsilon)}$.

---

[1] A recent work by Ganesh et al. (2025) argues for a broader definition of the Rashomon set, incorporating all decisions made during model development, including even data processing. Under this perspective, the different Rashomon sets across neighbouring datasets in our work can be seen as subsets of one larger Rashomon set. Although we do not adopt this perspective, since we compare data processing choices and their effects, it still offers a useful intuition to the reader for using a fixed threshold across neighbouring datasets.

**Proof Sketch.**  We first show that for neighbouring datasets, the Bayes optimal loss is proportional to the overlapping coefficient, under the assumption of identical class priors. Thus, we say that the overlapping coefficient is higher for the dataset with the higher Bayes optimal loss. We then use the loss relationship in the first assumption to show that any model in the Rashomon set of the higher-loss dataset also belongs to the Rashomon set of the lower-loss dataset, but not vice-versa, creating a subset relationship. Complete proof can be found in the Appendix (§A).

**Interpreting the Assumptions.**  The assumptions together state that one of the datapoints differing between neighbouring datasets is harder to classify than the other, and that all good models and both Bayes optimal models agree on this. The assumption fails when both differing datapoints lie in the ambiguous region near the decision boundary. A tighter Rashomon parameter $\epsilon$ (i.e., a smaller $\epsilon$) makes the ambiguous region smaller, increasing the likelihood that the assumption holds.

Note that if the Bayesian optimal models are in the Rashomon set, the second assumption becomes redundant. In other words, for any hypothesis class expressive enough to include the Bayesian optimal, the second assumption can be dropped.

## 4.2 EXTENDING TO $k$-NEIGHBOURING DATASETS

Our theoretical discussion has focused on $1$-neighbouring datasets, which enabled us to provide a rigorous proof for the downstream multiplicity based on the precise relationship between neighbouring datasets. However, in practice, we are unlikely to encounter datasets that differ by only a single data point. Instead, we typically face the more general and realistic case of $k$-neighbouring datasets. While our previous sets of proofs do not work directly in this setting, we propose the following conjecture:

**Conjecture 4.1.** Given two $k$-neighbouring binary classification datasets $D_{train}^1, D_{train}^2$ of size $n$, with $k \ll n$, if the overlapping coefficient between the two classes in higher for one dataset, i.e., without loss of generality $OVL_{train}^1 \geq OVL_{train}^2$, then the resulting multiplicity for this dataset under a common threshold $\epsilon$ will be a lower than the other dataset, i.e., $M_{\Theta^1} \leq M_{\Theta^2}$.

In addition to generalizing from $1$-neighbouring datasets to $k$-neighbouring datasets, we also shift our focus from the Rashomon set to the resulting multiplicity. Interestingly, the conjecture remains provable under strong assumptions—specifically, if the assumptions of Theorem 4.1 hold across all $k$ differing data points (see §A for details). However, as $k$ increases, such an assumption becomes increasingly unrealistic. Instead, we draw on our previous observations that a greater overlap between datasets is likely to increase the error across most models within the Rashomon set. As a result, given a fixed Rashomon parameter $\epsilon$, we expect lower multiplicity in datasets with higher overlap compared to their neighbours. This result is not provable, with corner cases where we might see opposite trends (for instance, monotonicity of a metric requires pure subset relationships, and multiplicity can increase in some cases even if the size of the Rashomon set is decreasing (Ganesh et al., 2025)). However, we expect our intuition derived from the theoretical analysis in Section 4.1 to hold in most real-world datasets, and we will support these claims through empirical evidence on two data processing tasks as case studies: data acquisition in active learning (§5) and data imputation (§6).

## 5 MULTIPLICITY AND ACTIVE LEARNING

With an understanding of how neighbouring datasets influence multiplicity, we extend our discussion to active learning. We empirically evaluate several data acquisition algorithms, alongside our own multiplicity-aware techniques. Our results reveal a negative correlation between the overlapping coefficient and the resulting multiplicity, as well as the success of our techniques in achieving the lowest (or highest) multiplicity without sacrificing accuracy.

## 5.1 NEIGHBOURING DATASETS IN ACTIVE LEARNING

In active learning, we have access to a large pool of unlabeled data, and the objective is to selectively acquire a small subset of the potentially most informative data points to be labeled, known as data acquisition. Typically, active learning begins with a small labeled dataset $D_{lab}^0$ and a large pool of unlabeled points $X_{unlab}^0$. At each timestep $t$, the algorithm uses the current labeled dataset $D_{lab}^t$ and the remaining unlabeled pool $X_{unlab}^t$ to select a batch of points $X_{or}^t \subset X_{unlab}^t$, to be labeled by the

| RF | \multicolumn Number of Steps $t$ | | | | | LR | Number of Steps $t$ | | | | | MLP | Number of Steps $t$ | | | | |
|---|---|---|---|---|---|---|---|---|---|---|---|---|---|---|---|---|---|
| | 1 | 2 | 3 | 4 | 5 | | 1 | 2 | 3 | 4 | 5 | | 1 | 2 | 3 | 4 | 5 |
| 500 | -0.21 | -0.34 | -0.12 | -0.01 | -0.02 | | -0.73 | -0.72 | -0.61 | -0.51 | -0.57 | | -0.54 | -0.35 | -0.31 | -0.38 | -0.25 |
| 1000 | -0.35 | -0.28 | -0.20 | -0.20 | -0.25 | | -0.84 | -0.78 | -0.61 | -0.52 | -0.44 | | -0.63 | -0.62 | -0.44 | -0.35 | -0.29 |
| 2000 | -0.41 | -0.31 | -0.27 | -0.22 | -0.19 | | -0.80 | -0.77 | -0.64 | -0.60 | -0.69 | | -0.85 | -0.65 | -0.48 | -0.43 | -0.28 |

Size of $D^0_{lab}$ ($n$)

Figure 3: Spearman's rank correlation coefficients between the overlap and resulting multiplicity.

oracle. Once labeled, these are added to the labeled dataset, i.e., $D^{t+1}_{lab} = D^t_{lab} + (X^t_{or}, Y^t_{or})$. We define the initial labeled set size $|D^0_{lab}| = n$, and $|X^t_{or}| = q$ points are labeled at each step.

Over a total of $T$ steps, two different active learning algorithms may choose distinct sequences of points to label. It is easy to see that the resulting labeled datasets can be considered $k$-neighbouring datasets with $k \leq Tq$. Thus, we argue that the choice between active learning strategies can also be seen as a choice between neighbouring datasets.

## 5.2 EXPERIMENT SETUP AND ALGORITHMS

Before jumping into the empirical results, we provide an overview of the experiment setup, as well as define our multiplicity-aware data acquisition algorithms.

**Dataset.** We use three different datasets, ACSIncome (Ding et al., 2021), ACSEmployment (Ding et al., 2021), and Bank Customer Churn dataset (Topre, 2025), to ensure the robustness of our findings. Due to limited space, we focus on the ACSIncome dataset in the main paper, while additional results and details of the experiment setup are delegated to the Appendix (§C).

We first divide the dataset into train and test sets, with a ratio of $[0.8, 0.2]$. Next, we sample $n$ points randomly from the train set that will serve as our $D^0_{lab}$, and test three different values of $n \in \{500, 1000, 2000\}$. The rest of the train set is our unlabeled pool of data. We run various active learning algorithms with a query size $q = 100$ for a total of $T = 5$ steps. The complete pipeline starting from sampling $D^0_{lab}$ is repeated 10 times, while sticking with the same test set.

**Models.** We use LogisticRegression (LR), RandomForest (RF), and Multi-Layer Perceptron (MLP) with a single hidden layer of size 10, three model classes of varying complexity. We use RF as our default setup, while additional results for LR and MLP are in the Appendix (§C). To evaluate multiplicity, for each dataset, we train a total of 100 models and then select the Rashomon set. As discussed during formalization (Definitions 3.1, 3.2), the creation of the Rashomon set is done using model loss on the train set, while all evaluations are performed on the test set.

**Evaluation Metrics.** We use accuracy (0-1 scale) as a performance measure and ambiguity (Definition 3.2) as a measure of multiplicity. We use histogram-based binning to approximate the underlying class-conditional probability distributions to calculate the overlapping coefficient. More details are delegated to the Appendix (§C).

**Baseline Algorithms and Multiplicity-Aware Data Acquisition.** We study three common baselines for active learning: (a) Random (Aggarwal et al., 2014), data points to be labeled are chosen at random, (b) Confidence (Aggarwal et al., 2014), data points with the lowest prediction confidence are chosen, and (c) Committee (Seung et al., 1992), data points with the most conflicting predictions from a committee of 100 models trained on the current labeled data are chosen. Prediction confidence here refers to the output probability of the model's prediction.

In addition, we propose two new data acquisition algorithms: (a) MultLow, which trains a committee of models on the labeled data and chooses the data points with low confidence in all models of the committee, and (b) MultHigh, which is similar but instead chooses the data points with high confidence in all models of the committee. Pseudocode for both algorithms is in the Appendix (§B).

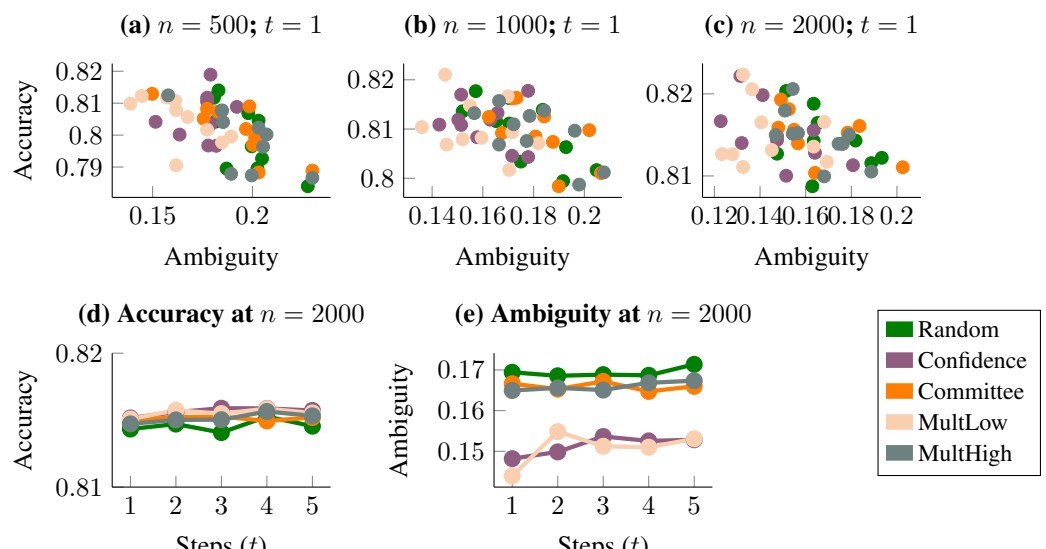

Figure 4: **(a, b, c)** Accuracy and ambiguity across various strategies for one step of data acquisition. We see clear trends of our MultLow (and MultHigh) approach(es) getting the lowest (and highest) multiplicity consistently, while maintaining similar accuracies. **(d, e)** Accuracy and ambiguity across multiple steps of data acquisition. Similar trends persist across multiple steps of active learning.

## 5.3 CONTROLLING MULTIPLICITY DURING ACTIVE LEARNING

We start by examining the relationship between the overlapping coefficient and multiplicity for varying initial labeled sizes ($n$) and active learning steps ($t$), across all algorithms. Average correlation scores across all random seeds are reported in Figure 3 (standard deviations are present in the Appendix). We see a clear negative correlation on average, supporting our hypothesis that higher overlap leads to lower multiplicity (Conjecture 4.1). Unsurprisingly, the correlation is stronger when $n$ is large or $t$ is small, i.e., settings where $k \ll n$. Moreover, we notice the correlations also depend on the choice of the learning algorithm, stronger for LR and MLP compared to RF.

Moving beyond the overall correlation, we next analyze the trends exhibited by each algorithm separately in Figure 4. Our algorithms, MultLow and MultHigh, are consistently among the techniques achieving the lowest (and highest) multiplicity. Even in scenarios where our theoretical assumptions do not hold—such as when $n$ is small or $t$ is large—the efficacy of our algorithms, MultLow and MultHigh, indicates that our insights extend well beyond strict theoretical settings. This robustness highlights the practical utility of our approach across a broader range of real-world scenarios involving neighbouring datasets.

Note that MultHigh achieves similar multiplicity as a few other baseline techniques, but does not surpass them. This is not surprising, as there might be limits to how much uncertainty is present in the data, as well as limits to the approximations in our algorithm design. While MultHigh (and corresponding MultLow) can surpass existing techniques in some scenarios (see Figures 8, 10, 12 in the appendix), this isn't always necessary. Instead, what these techniques give us is control over steering towards increasing or decreasing multiplicity while maintaining performance.

## 6 MULTIPLICITY AND DATA IMPUTATION

We now turn to our second application: data imputation, repeating the $k$-neighbouring dataset formulation and the empirical study on existing algorithms alongside our own multiplicity-aware techniques. Combined with the results from active learning, these studies underscore the practical utility of our framework in analyzing and guiding developer decisions during data processing.

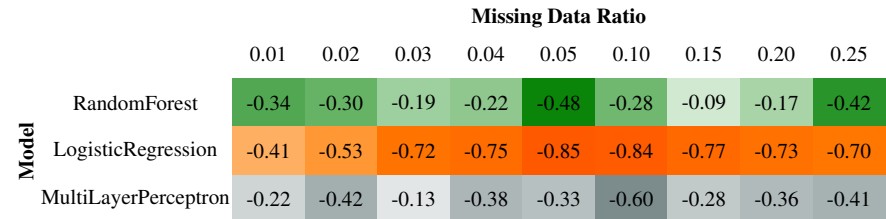

Figure 5: Correlation between the overlapping coefficient and resulting multiplicity.

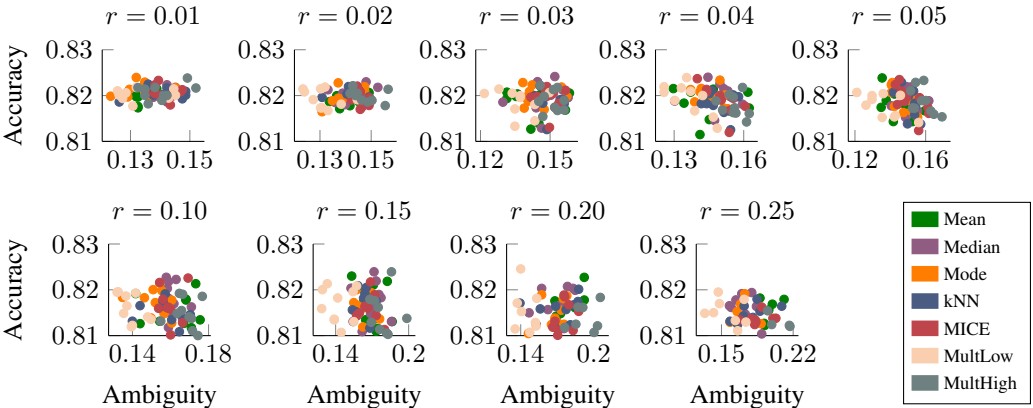

Figure 6: Accuracy and ambiguity for various data imputation strategies across varying values of missing data ratio $r$. MultLow (and MultHigh) algorithms stand out more for higher values of missing data ratio $r$, highlighting that a large amount of missing data can make the imputation more steerable.

## 6.1 NEIGHBOURING DATASETS IN DATA IMPUTATION

Data imputation fills the missing values in a dataset to best reflect what the real values might have been. It is a necessary step before learning, as most models cannot handle data with missing values (Miao et al., 2022). Given a dataset $D^{mis}$ with missing values $S^{mis} = \{ij|D_{ij}^{mis} = \phi\}$, a data imputation algorithm fills them with a set of non-empty values $S^{imp} = \{s_{ij}|ij \in S^{mis} \text{ and } s_{ij} \neq \phi\}$. The final imputed dataset can be defined as $D^{imp} = D^{mis} \oplus S^{imp}$, where the $\oplus$ operator represent filling the values missing in $D^{mis}$ with values from $S^{imp}$. We define $|D^{mis}| = n$ and $|S^{mis}| = s$.

Two different imputation techniques may fill the missing values in different ways while the rest of the dataset remains unchanged, and the resulting imputed dataset will be $k$-neighbouring datasets, where $k \leq s$. Similar to active learning, we argue that the choice between data imputation techniques can also be seen as a choice between neighbouring datasets, thus fitting within our broader discussion.

## 6.2 EXPERIMENT SETUP AND ALGORITHMS

We use the same experiment setup as before, but with the following differences (more details in §D),

**Dataset.** After dividing the dataset into train and test sets, we randomly remove $r$ fraction of values from the train set, giving us our $D^{miss}$. The complete pipeline is repeated 10 times.

**Baseline Algorithms and Multiplicity-Aware Data Imputation.** We study five commonly used baselines in imputation: (a) Mean (Miao et al., 2022), filling with the mean of the feature, (b) Median (Miao et al., 2022), filling with the median of the feature, (c) Mode (Miao et al., 2022), filling with the mode of the feature, (d) kNN (Altman, 1992), using k-nearest neighbours algorithm to find 5 neighbours and fill with the mean of their value, and (e) MICE (Van Buuren & Groothuis-Oudshoorn, 2011), learning predictors of a feature using other features, one at a time, and improving iteratively.

In addition, our multiplicity-aware imputation techniques include: (a) MultLow, which checks the confidence of the data point for all five baseline imputations and chooses the one with the least confidence, repeating for all missing values, and (b) MultHigh, which instead chooses the one with the highest confidence. To get the confidence scores, we train a single model on the mean-imputed dataset. Our multiplicity-aware imputation algorithms use existing imputation techniques and choose between them for every missing value. We provide pseudocode in the Appendix (§B).

### 6.3 STRONGER CONTROL WITH MORE MISSING VALUES

We start with the relationship between overlapping coefficients and the resulting multiplicity for data imputation, in Figure 5. As expected, we observe negative correlations, particularly at smaller missing value ratios where the neighbouring dataset assumption holds.

The most intriguing results, however, come from our multiplicity-aware algorithms. In Figure 6, we present the average accuracy and resulting multiplicity across all random seeds and imputation techniques, evaluated over varying levels of missing data. Our techniques consistently achieve the lowest (or highest) multiplicity, but what stands out is that these trends become more pronounced at higher missing value ratios. With more missing data, the number of plausible imputations—and thus neighbouring datasets—increases. This leads to many neighbouring datasets varying substantially in downstream multiplicity, making our multiplicity-aware methods more valuable.

### 7 CONCLUSION AND FUTURE WORK

In this work, we introduced a neighbouring datasets framework to study the impact of data processing on multiplicity, offering a practical lens on the interplay between dataset and multiplicity. Our framework captures a wide range of data processing scenarios, provides theoretical insights into the relationship between neighbouring datasets and multiplicity, and reveals a surprising trend supported by rigorous proofs. We also demonstrated its utility through active learning and data imputation. Our method enables practitioners to anticipate downstream multiplicity without the need to explicitly train multiple models. This avoids the prohibitive computational cost and practical infeasibility of training the large number of models required to approximate a Rashomon set for every single data processing decision. Instead, our framework provides a principled way to assess how data-processing decisions influence multiplicity upfront.

Moreover, we introduce multiplicity-aware strategies for active learning and data imputation. These methods match the computational efficiency of standard baselines while leveraging our framework to intentionally steer the process toward higher or lower multiplicity. . While the debate on whether to increase or decrease multiplicity is ongoing, our work provides tools to study the impact of data processing choices on downstream multiplicity and can help developers make informed choices in either case.

Looking ahead, an important avenue for future research is establishing a formal connection between our neighbouring dataset framework and differential privacy, which could yield valuable theoretical and practical insights in the future. Another promising direction involves revisiting the definition of neighbouring datasets, as alternatives based on $L_1/L_2$ distances may offer a closer alignment with robustness literature. This perspective opens up opportunities to study the influence of distribution shifts and adversarial data on multiplicity through the same lens of neighbouring datasets.

Moreover, our work has focused explicitly on binary classification, using the overlapping coefficient to measure the 'difficulty' of a dataset, similar to noise in the data by Semenova et al. (2024) or separability by Watson-Daniels et al. (2023b). However, as we move to problems beyond binary classification, the measure of the 'difficulty' of a dataset gets complicated (Ethayarajh et al., 2022). We expect the intuitions behind our framework to remain, i.e., if data processing choices increase the 'difficulty' of a dataset, the error for all models is likely to increase, resulting in lower multiplicity. Yet, this extension to problems beyond binary classification is non-trivial, representing another important direction of future research.

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

## A    PROOF FOR THEOREM 4.1 AND INSIGHTS FOR CONJECTURE 4.1

**Step 1: Bayes optimal 0-1 loss in terms of Overlapping Coefficient**    The Bayes optimal classifier minimizes the 0-1 loss, predicting the class with the higher posterior probability at each $x$. So the Bayes classifier $f_{\theta^*}(x)$ predicts:

$$f_{\theta^*}(x) = \arg \max_{y \in \{0,1\}} P_y(x).$$

Thus, the Bayes 0-1 loss $L^*$ can be expressed as the expected probability of misclassification:

$$L^* = \mathbb{E}_x \left[ \min(P_0(x), P_1(x)) \right] = \int_x \min(\pi_0 P_0(x), \pi_1 P_1(x)).$$

where $\pi_0, \pi_1$ are class priors for both classes. Assuming identical class priors, we can simplify it as,

$$L^* = \frac{1}{2} \int_x \min(P_0(x), P_1(x)) = \frac{1}{2} OVL(P_0, P_1).$$

Hence, for the two neighbouring datasets:

$$L_1^* = \frac{1}{2} OVL_{\text{train}}^1, \quad L_2^* = \frac{1}{2} OVL_{\text{train}}^2.$$

Therefore, if $L_1^* \geq L_2^*$, then:

$$OVL_{\text{train}}^1 \geq OVL_{\text{train}}^2.$$

**Step 2: Overlapping coefficient is higher for $D_{\text{train}}^1$**    We know from our second assumption that:

$$L(\theta_1^*, (x_0^1, y_0^1)) \geq L(\theta_2^*, (x_0^2, y_0^2)).$$

Since all other training examples are shared between the two datasets, the only difference in their total empirical losses lies in this one datapoint. Thus, the empirical loss satisfies:

$$L_1^* \geq L_2^*,$$

From the derivation in Step 1, we thus get:

$$OVL_{\text{train}}^1 \geq OVL_{\text{train}}^2.$$

**Step 3: Subset relationship from loss dominance**    Now let $\theta \in \Theta_{(D_{\text{train}}^1, \epsilon)}$, i.e., $L_{D_{\text{train}}^1}(\theta) \leq \epsilon$. Since $D_{\text{train}}^1$ and $D_{\text{train}}^2$ differ in only one datapoint, we can write:

$$L_{D_{\text{train}}^1}(\theta) = \frac{1}{n} \left( L(\theta, (x_0^1, y_0^1)) + \sum_{j=1}^{n-1} L(\theta, (x_j, y_j)) \right),$$

$$L_{D_{\text{train}}^2}(\theta) = \frac{1}{n} \left( L(\theta, (x_0^2, y_0^2)) + \sum_{j=1}^{n-1} L(\theta, (x_j, y_j)) \right).$$

Subtracting:

$$L_{D_{\text{train}}^1}(\theta) - L_{D_{\text{train}}^2}(\theta) = \frac{1}{n} \left( L(\theta, (x_0^1, y_0^1)) - L(\theta, (x_0^2, y_0^2)) \right) \geq 0,$$

by the assumed loss inequality in our first assumption. Therefore:

$$L_{D_{\text{train}}^2}(\theta) \leq L_{D_{\text{train}}^1}(\theta) \leq \epsilon \quad \Rightarrow \quad \theta \in \Theta_{(D_{\text{train}}^2, \epsilon)}.$$

Since this holds for all $\theta \in \Theta_{(D_{\text{train}}^1, \epsilon)}$, but not necessarily the other way around, we conclude:

$$\Theta_{(D_{\text{train}}^1, \epsilon)} \subseteq \Theta_{(D_{\text{train}}^2, \epsilon)}.$$

## A.1   EXTENSION TO $k$-NEIGHBOURING DATASETS

Let $D_{\text{train}}^1$ and $D_{\text{train}}^2$ be $k$-neighbouring binary classification datasets, i.e., they differ at $k$ indices $\mathcal{I} = \{i_1, \ldots, i_k\}$, such that for all $j \in \mathcal{I}$, we have:

$$(x_j^1, y_j^1) \neq (x_j^2, y_j^2),$$

and for all other $j \notin \mathcal{I}$, the examples are shared:

$$(x_j^1, y_j^1) = (x_j^2, y_j^2).$$

Suppose further that the per-point loss dominance condition holds at all differing indices:

$$L(\theta, (x_j^1, y_j^1)) \geq L(\theta, (x_j^2, y_j^2)) \quad \forall \theta \in \Theta_{(D_{\text{train}}^1, \epsilon)} \cup \Theta_{(D_{\text{train}}^2, \epsilon)}, \quad \forall j \in \mathcal{I}.$$

and

$$L(\theta_1^*, (x_j^1, y_j^1)) \geq L(\theta_2^*, (x_j^2, y_j^2)) \quad \forall j \in \mathcal{I}.$$

Then the overlapping coefficient between the classes is higher for $D_{\text{train}}^1$, i.e.,

$$OVL_{\text{train}}^1 \geq OVL_{\text{train}}^2,$$

and the Rashomon set satisfies:

$$\Theta_{(D_{\text{train}}^1, \epsilon)} \subseteq \Theta_{(D_{\text{train}}^2, \epsilon)}.$$

To prove this, we can simply decompose the $k$-neighbouring datasets into a sequence of $k$ consecutive 1-neighbouring transitions:

$$D_{\text{train}}^1 = D^{(0)} \to D^{(1)} \to \cdots \to D^{(k)} = D_{\text{train}}^2,$$

where each $D^{(t)}$ and $D^{(t+1)}$ differ at exactly one datapoint $(x_t, y_t)$, and the loss dominance condition holds at each step. From Theorem 4.1, each such one-step transition satisfies:

$$\Theta_{(D^{(t)}, \epsilon)} \subseteq \Theta_{(D^{(t+1)}, \epsilon)}.$$

Applying this sequentially:

$$\Theta_{(D_{\text{train}}^1, \epsilon)} = \Theta_{(D^{(0)}, \epsilon)} \subseteq \Theta_{(D^{(1)}, \epsilon)} \subseteq \cdots \subseteq \Theta_{(D^{(k)}, \epsilon)} = \Theta_{(D_{\text{train}}^2, \epsilon)}.$$

Thus,

$$\Theta_{(D_{\text{train}}^1, \epsilon)} \subseteq \Theta_{(D_{\text{train}}^2, \epsilon)}.$$

# B PSEUDOCODE FOR ALGORITHMS

## B.1 MULTLOW AND MULTHIGH FOR DATA ACQUISITION

---
**Algorithm 1** MultLow for Data Acquisition
---
**Require:** Labeled dataset $L$, unlabeled dataset $U$, query size $Q$, committee size $K$
1: Train a committee of $K$ models $\{M_1, M_2, \ldots, M_K\}$ on $L$
2: Initialize $S \leftarrow \emptyset$
3: **for** each $x \in U$ **do**
4:    Compute confidence $c_i(x)$ from each model $M_i$
5:    Compute maximum confidence across all models: $c_{\max}(x) = \max_i c_i(x)$
6: **end for**
7: Select bottom-$Q$ points with lowest $c_{\max}(x)$ values
8: $S \leftarrow$ selected points
9: **return** $S$

---

---
**Algorithm 2** MultHigh for Data Acquisition
---
**Require:** Labeled dataset $L$, unlabeled dataset $U$, query size $Q$, committee size $K$
1: Train a committee of $K$ models $\{M_1, M_2, \ldots, M_K\}$ on $L$
2: Initialize $S \leftarrow \emptyset$
3: **for** each $x \in U$ **do**
4:    Compute confidence $c_i(x)$ from each model $M_i$
5:    Compute minimum confidence across all models: $c_{\min}(x) = \min_i c_i(x)$
6: **end for**
7: Select top-$Q$ points with highest $c_{\min}(x)$ values
8: $S \leftarrow$ selected points
9: **return** $S$

---

## B.2 MULTLOW AND MULTHIGH FOR DATA IMPUTATION

---

**Algorithm 3** MultLow for Data Imputation

---

**Require:** Dataset with missing values $D$, set of baseline imputations $\{I_{mean}, I_{median}, \ldots, I_{mice}\}$
1: Train a model $C$ on mean-imputed version of $D$
2: Initialize $D' \leftarrow D$
3: **for** each record $r$ with missing values in $D$ **do**
4:     **for** each imputation method $I_j$ **do**
5:        Compute imputed record $r_j$ using $I_j$
6:        Compute confidence score $c_j = C(r_j)$
7:     **end for**
8:     Select $r^* = r_j$ with lowest confidence $c_j$
9:     Fill $r$ in $D'$ with $r^*$
10: **end for**
11: **return** $D'$

---

**Algorithm 4** MultHigh for Data Imputation

---

**Require:** Dataset with missing values $D$, set of baseline imputations $\{I_{mean}, I_{median}, \ldots, I_{mice}\}$
1: Train a model $C$ on mean-imputed version of $D$
2: Initialize $D' \leftarrow D$
3: **for** each record $r$ with missing values in $D$ **do**
4:     **for** each imputation method $I_j$ **do**
5:        Compute imputed record $r_j$ using $I_j$
6:        Compute confidence score $c_j = C(r_j)$
7:     **end for**
8:     Select $r^* = r_j$ with highest confidence $c_j$
9:     Fill $r$ in $D'$ with $r^*$
10: **end for**
11: **return** $D'$

---

## C ADDITIONAL RESULTS FOR ACTIVE LEARNING

In this appendix section, we provide detailed results across all datasets for active learning. We find similar trends across various datasets and models as seen in the main paper.

### C.1 EXPERIMENT SETUP DETAILS

**Folktables Subset.** We use the "New Mexico" state subset for both ACSIncome and ACSEmployment throughout the paper.

**Choosing Rashomon parameter $\epsilon$.** To make sure the Rashomon set contains enough models for each algorithm in our setup while keeping the threshold tight, the value of $\epsilon$ is chosen to be the smallest value possible such that there are at least 50 models in the Rashomon set for each setup. The $\epsilon$ value is chosen separately for each setting, i.e., each random seed, initial dataset size, and number of steps; but is shared between all different algorithms, i.e., a common Rashomon parameter $\epsilon$ is used across algorithms for any particular setting.

**Estimating Probability Density.** We use histogram-based binning for density estimation, a standard and well-established approach, before we calculate the overlapping coefficient for each dataset. We create 5 bins across each feature (resulting in a total of $number\_of\_features * 5$ bins) for approximating the probability density. For each dataset, the bins are first created based on the complete dataset with both classes, and then class-conditional probability distributions are estimated using class-specific data subsets, to make sure the two distributions have the same set of bins.

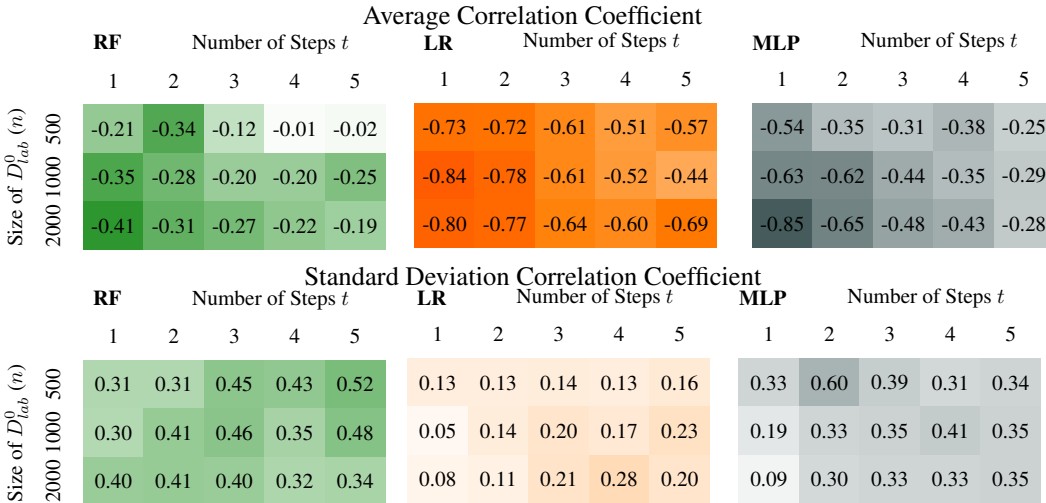

Figure 7: Average and standard deviation of spearman's rank correlation coefficients between the overlap and resulting multiplicity for the ACSIncome dataset.

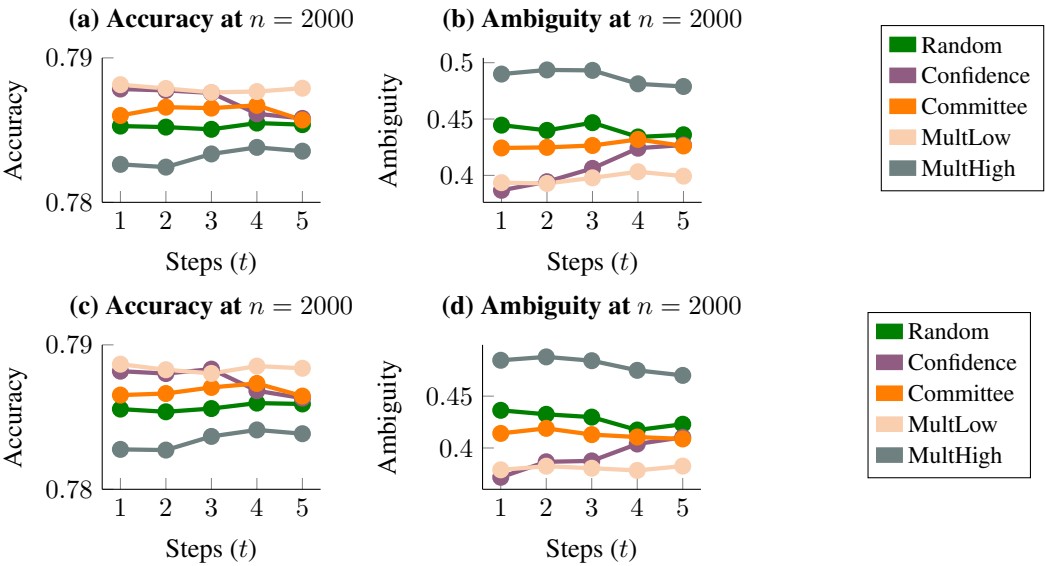

Figure 8: Accuracy and ambiguity across multiple steps of data acquisition for LR (top, (a), (b)) and MLP (bottom, (c), (d)) models for ACSIncome dataset. Similar trends persist across multiple steps of active learning.

## C.2    ALL RESULTS FOR ACSINCOME DATASET

Here, we provide detailed results for active learning on the ACSIncome dataset. First, we restate the results in Figure 3, along with the standard deviations recorded separately, present in Figure 7. We then repeat the experiments in Figure 4(d, e) for LR and MLP models, and the results are presented in Figure 8.

## C.3    ALL RESULTS FOR ACSEMPLOYMENT DATASET

Here, we provide detailed results for active learning on the ACSEmployment dataset. First, we repeat the experiments in Figure 3, along with the standard deviations recorded separately, present in Figure

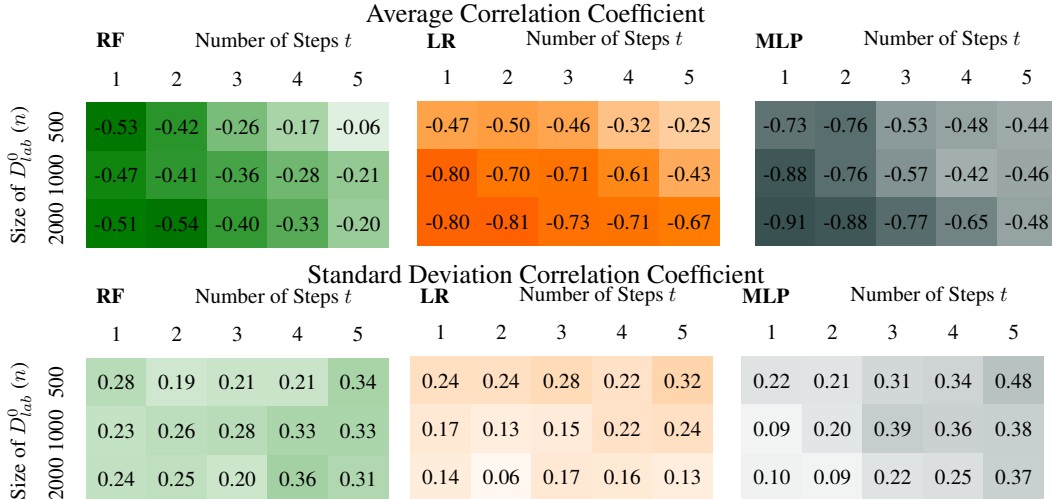

Figure 9: Average and standard deviation of spearman's rank correlation coefficients between the overlap and resulting multiplicity for the ACSEmployment dataset.

9. We then repeat the experiments in Figure 4(d, e) for all three model types, and the results are presented in Figure 10.

### C.4 ALL RESULTS FOR BANK DATASET

Here, we provide detailed results for active learning on the Bank dataset. First, we repeat the experiments in Figure 3, along with the standard deviations recorded separately, present in Figure 11. We then repeat the experiments in Figure 4(d, e) for all three model types, and the results are presented in Figure 12.

## D ADDITIONAL RESULTS FOR DATA IMPUTATION

In this appendix section, we provide detailed results across all datasets for data imputation. We find similar trends across various datasets and models as seen in the main paper.

### D.1 EXPERIMENT SETUP DETAILS

**Folktables Subset.** Same details as above in §C.1

**Choosing Rashomon parameter $\epsilon$.** Same details as above in §C.1.

### D.2 ALL RESULTS FOR BANK DATASET

Here, we first repeat the experiments in Figure 5 for the Bank dataset and provide the results in Figure 13. Next, we repeat the experiments in Figure 6 for the Bank dataset using RandomForests, and the results are presented in Figure 14.

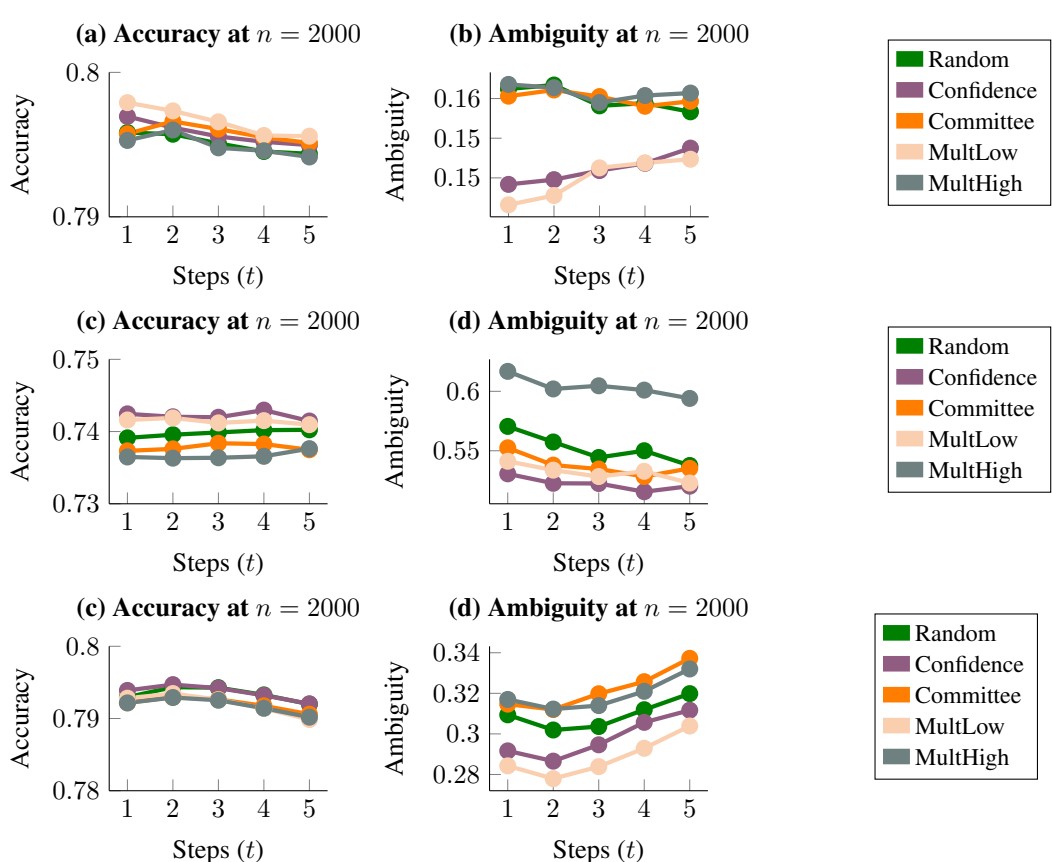

Figure 10: Accuracy and ambiguity across multiple steps of data acquisition for RF (top, (a), (b)), LR (middle, (c), (d)) and MLP (bottom, (e), (f)) models for ACSEmployment dataset. Similar trends persist across multiple steps of active learning.

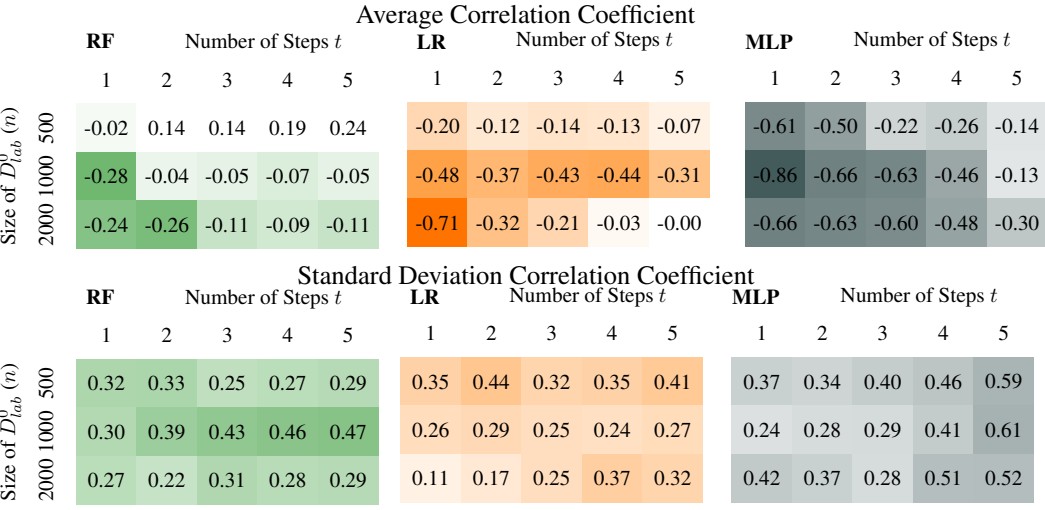

Figure 11: Average and standard deviation of spearman's rank correlation coefficients between the overlap and resulting multiplicity for the Bank dataset.

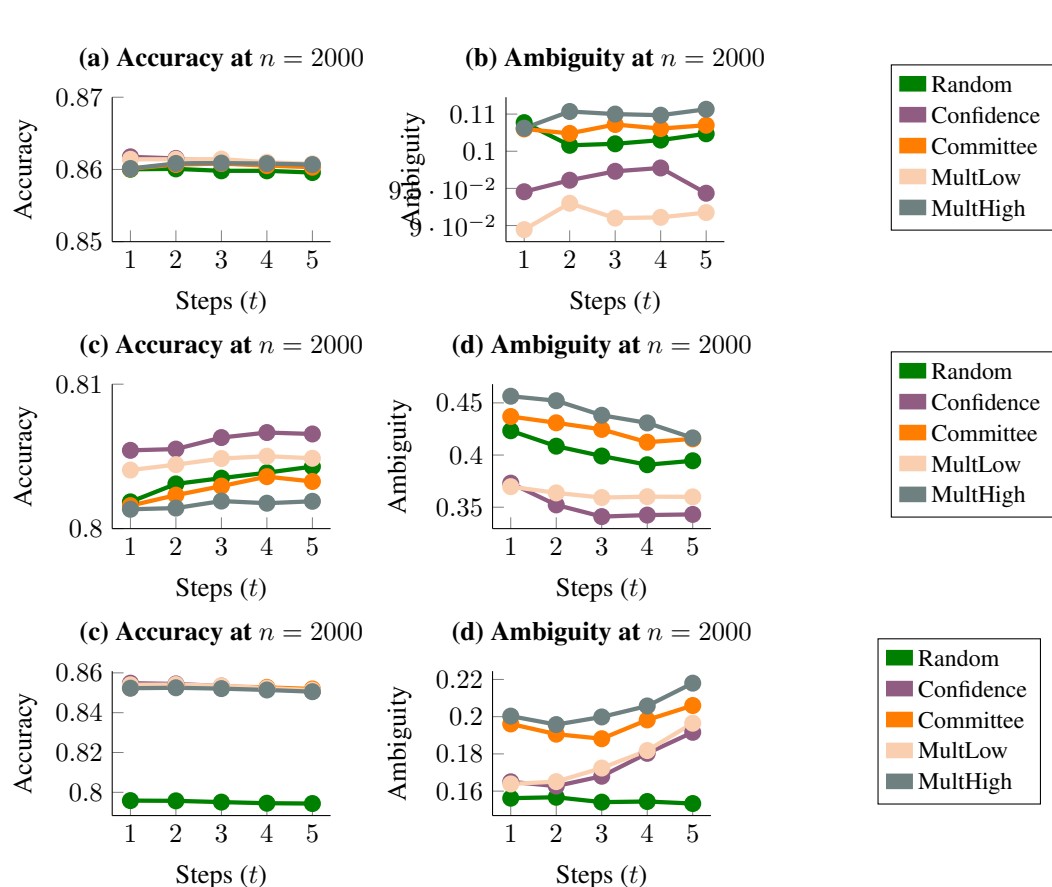

Figure 12: Accuracy and ambiguity across multiple steps of data acquisition for RF (top, (a), (b)), LR (middle, (c), (d)) and MLP (bottom, (e), (f)) models for Bank dataset. Similar trends persist across multiple steps of active learning.

| | | Missing Data Ratio | | | | | | | | |
|---|---|---|---|---|---|---|---|---|---|---|
| | | 0.01 | 0.02 | 0.03 | 0.04 | 0.05 | 0.10 | 0.15 | 0.20 | 0.25 |
| **Model** | RandomForest | -0.50 | -0.49 | -0.37 | -0.40 | -0.29 | -0.01 | 0.17 | 0.32 | 0.41 |
| | LogisticRegression | 0.17 | -0.21 | -0.27 | -0.44 | -0.69 | -0.56 | -0.57 | -0.49 | -0.16 |
| | MultiLayerPerceptron | -0.26 | -0.21 | -0.08 | -0.19 | -0.18 | -0.12 | -0.05 | -0.17 | -0.44 |

Figure 13: Correlation between the overlapping coefficient and resulting multiplicity for the Bank dataset.

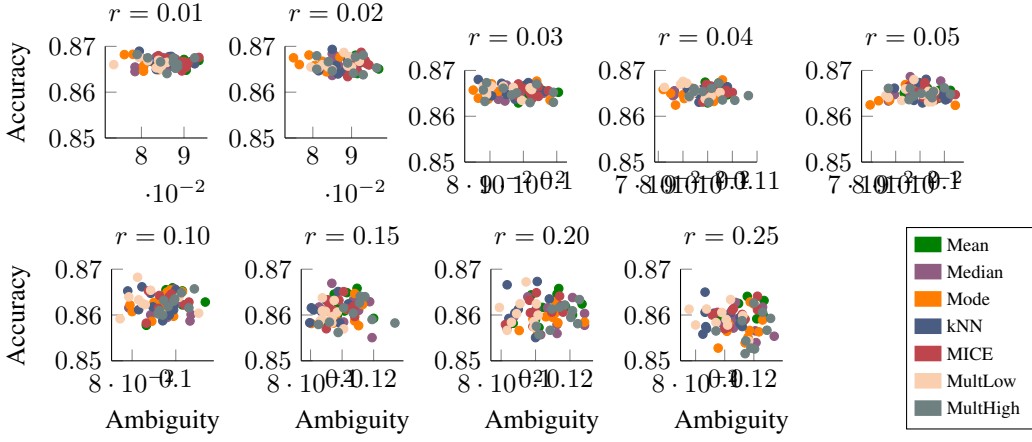

Figure 14: Accuracy and ambiguity for various data imputation strategies across varying values of missing data ratio $r$ for Bank dataset.

