# OpenReview forum: "Data as a Lever: A Neighbouring Datasets Perspective on Predictive Multiplicity"
_ICLR.cc/2026/Conference — Submitted to ICLR 2026_

### Official Review · Reviewer_xCmH · 2025-10-29

**Soundness:** 2
**Presentation:** 4
**Contribution:** 2
**Rating:** 4
**Confidence:** 3

**Summary:**

The paper investigates data-preprocessing-induced multiplicity, i.e. the existence of multiple “good enough” models that arise from different versions of a dataset in classification settings.  It introduces the notion of $k-$neighbouring datasets, i.e., datasets containing the same number of samples $n$ but differing in $k$ of them.  Such datasets can typically be obtained by applying different preprocessing techniques (eg different imputation techniques).

The authors build on two key notions:
- **Rashomon set**: the set of models achieving at most a given loss on the training set ($\approx$ the set of good enough models)
- **Multiplicity** (taken as ambiguity in the paper): the proportion of test samples on which at least two models from the Rashomon set disagree (this is a score between 0 and 1).

The paper proves **a theorem showing that, for 1-neighbouring datasets, greater class overlap leads to a smaller Rashomon set**; in other words, when the distributions of the two classes overlap more, the set of “good enough” models shrinks.

Motivated by this result, the authors formulate **a conjecture: for two $k$-neighbouring datasets, higher class overlap implies lower multiplicity.**

Experiments creating $k-$neighbouring datasets via active learning or imputation empirically confirm that higher overlap coefficients are negatively correlated with multiplicity.

Finally, the paper proposes and evaluates **active learning and imputation strategies designed to achieve comparable accuracy while inducing either lower or higher multiplicity** than standard baselines.

**Strengths:**

I found the paper particularly clear, which is noteworthy given the number of notions introduced. I also appreciated the explicit discussion of assumptions and the careful formulation of the conjecture, reflecting a concern for transparency rather than an attempt to overstate the results.

The related work is thoroughly and appropriately covered.

The authors propose an original perspective on multiplicity by jointly considering the effects induced by both datasets and models. Interestingly, their findings reverse some previous conclusions, highlighting how different views can lead to contrasting outcomes.

**Weaknesses:**

* **My main concerns relates to the multiplicity-aware data acquisition and imputation strategies. These are designed to either minimise or maximise multiplicity. However, I wonder whether it makes sense to minimize or maximize multiplicity.** In my understanding, multiplicity is a diagnostic rather than an objective. Minimising multiplicity artificially suppresses legitimate uncertainty. Creating an imputation technique that leads to small downstream multiplicity means I am drawing robust conclusions *given the choice of imputation*, but for the same samples, conclusions would not be robust under different imputations. Could the authors elaborate on the practical usefulness of the proposed multiplicity-aware data acquisition and imputation strategies?

* The authors hypothesise that smaller Rashomon sets lead to smaller multiplicity. This is a core hypothesis that leads to the proposed conjecture. If I understand correctly, the conjecture follows the following reasoning: more overlap => (1) more errors => (2) less models in the Rashomon set for a fixed threshold => (3) less multiplicity because less models mean less opportunities to have disagreement. Is this correct? Actually I am not convinced by transition (3) for multiplicity, as smaller Rashomon sets do not necessarily imply smaller multiplicity. Could the authors elaborate on it? The experiments indeed show that higher overlap (and so smaller Rashomon sets) lead to smaller multiplicity, but should we expect it to always hold?

* In the experiments, I agree that the proposed methods for low multiplicity (MultLow) indeed achieves lower multiplicity at a similar accuracy (although Figure 4e indicates that it is on par with the Confidence method in terms of multiplicity). However, I do not see in the Figures that MultHigh achieves higher multiplicity than the baselines.

**Questions:**

see weaknesses

---

> ### Author Response · Authors · 2025-11-22
>
> We are glad the reviewer found our work particularly clear and our results interesting. We appreciate the questions raised by the reviewer, and address them below. Our discussions here have also been incorporated to create the revised version of the paper now updated on Openreview, with revised parts marked in color blue.
>
> > Why control multiplicity?
>
> The choice to reduce or increase multiplicity is one that requires significant debate. We briefly distill the current arguments from the literature, but we also direct the reviewer to Gur-Arieh and Lee (2025) - https://dl.acm.org/doi/10.1145/3715275.3732215
>
> When dealing with applications that are normative or have multiple actors, we want to maximize multiplicity. For example, consider resume screening. Here, having higher multiplicity and diversity is useful, and data processing choices that might block certain possibilities can create blanket rejections for some individuals. On the other hand, when working with applications that are factual or have only a single actor, it is instead preferred to reduce multiplicity, even though sometimes this might be built on an arbitrary choice (or what Gur-Arieh and Lee (2025) call ‘arbitrarily consistent’). For instance, consider medical scenarios, where an important factor for continued use of these systems is the trust in them. If their outcomes change arbitrarily between models, individuals may lose trust in the system.
>
> We also want to emphasize that our work is valuable beyond the debate on whether to increase or decrease multiplicity. We provide tools to study the impact of data processing choices on downstream multiplicity, which would play a vital role in helping developers make informed choices. For instance, even if the reviewer believes that multiplicity should only be maximized (i.e., certain paths should not be blocked by developer decisions during data processing), our work is still valuable, as it can help developers avoid data processing choices that might restrict the downstream multiplicity.
>
> We have now added these discussions in the Introduction (Lines 39-45) and Conclusion (Lines 523-526), in the revised version of the paper.
>
> > Do smaller Rashomon sets imply less multiplicity?
>
> Our intuition that smaller Rashomon sets result in lower multiplicity comes from a property of multiplicity metrics called ‘monotonicity’ defined by Ganesh et al. (2025). Many metrics for multiplicity are ‘monotonic within the Rashomon set’, i.e., will provably increase or stay the same if models are added to the Rashomon set.
>
> Consider, for instance, ambiguity, a very popular multiplicity metric (Definition 3.2). Ambiguity measures the percentage of data points whose predictions can change due to the choice of the model. Adding more models to the Rashomon set does not affect the data points whose predictions could already change in the previous set of models, but can only make new data points also potentially change predictions, and thus ambiguity only increases or stays the same. Ganesh et al. (2025) provides an extensive list of multiplicity metrics, and which of them are monotonic.
>
> Note that ‘monotonicity’ relies on the subset relationship between two Rashomon sets, and not just size. Hence, it is not necessary that a reduction in the size of the Rashomon set would always result in lower multiplicity. This is why we only talk about subset relationships when discussing our proofs, and move to multiplicity when we are defining our conjecture. However, we believe that the trend can still be expected in most real world cases, hence our hypothesis, which is later supported with empirical evidence. We have now added further details about ‘monotonicity’ (Lines 224-226) and added these clarifications about our intuition (Lines 304-308) in the revised version of the paper.
>
> > MultHigh does not achieve higher multiplicity
>
> Results for both MultLow and MultHigh techniques are always robustly among the lowest (and correspondingly highest) multiplicity achieving techniques.
>
> Perhaps the reviewer is expecting that MultHigh (or similarly MultLow) should not only be among the highest (or lowest) multiplicity, but be even higher (or lower) than any existing algorithm. This isn’t always necessary, as there might be limits to how much uncertainty is present in the data, and the approximations in our algorithm design. It can still happen, and the reviewer can find more extreme scenarios in the Appendix, in Figures 8, 10, and 12, where MultHigh surpasses existing techniques and achieves notably higher multiplicity on some datasets.
>
> However, we emphasize again that these techniques might not always go beyond the multiplicity range captured by existing techniques, but only that they help us control and steer towards increasing or decreasing multiplicity while maintaining performance. We have now added this clarification explicitly during the discussion of results in the revised version of the paper (Lines 419-424).

---

### Official Review · Reviewer_HugR · 2025-10-29

**Soundness:** 2
**Presentation:** 1
**Contribution:** 2
**Rating:** 2
**Confidence:** 3

**Summary:**

This paper studies multiplicity, the phenomenon where multiple distinct models achieve similar performance. While previous research has focused mainly on model design, this work highlights the role of data in shaping multiplicity. The authors propose a neighboring datasets framework to analyze how changing a single data point affects model multiplicity. Surprisingly, they find that datasets with greater inter-class overlap show lower multiplicity, a result explained through a shared Rashomon parameter.
They further apply this framework to active learning and data imputation, conducting an analysis of multiplicity in these areas and introducing multiplicity-aware methods for both tasks.

**Strengths:**

This paper addresses an interesting problem, and practical problem encountered by a lot of practitioners.

**Weaknesses:**

The presentation currently lacks rigor and precision.

The theoretical framework would benefit from clearer definitions, more consistent notation, and stronger connections to related literature.

Given that the authors claim their method to be efficient, it should be described in a much clearer and more structured manner, with each step of the procedure explicitly outlined and all underlying concepts rigorously defined.

**Questions:**

*1. Lack of rigor and clarity in definitions.*
The manuscript would benefit from a clearer distinction between model parameters and the prediction functions parameterized by these parameters.
The current notation blurs this distinction and leads to confusion.

*2. Related work and conceptual connections.* The notion of leverage point is well defined in the context of linear regression, and this connection should be explicitly discussed if relevant.
Similarly, the relationship to stability-based approaches is not mentioned but seems important.
A short discussion would help situate the paper within the existing literature.

*3. Need for precise and rigorous statements.*
Several notations and definitions lack clarity:

The notation $0^i_{\mathrm{train}}$ is unclear and should be revised.

The definition of $OVL^1_{\mathrm{train}}$ is missing.
Please specify which probability density is used in this computation.
Is it based on a theoretical density, or on the empirical distribution associated with the training set?

*4. Interpretation of correlations (Line 363).*
The statement “Correlations are stronger for LR and MLP, which may be attributed to a poorer approximation of the model than RF” seems incorrect.
A more plausible explanation is that correlations measure linear dependencies, which are more naturally captured by linear regression (LR) and multilayer perceptrons (MLP) than by random forests (RF).

*5. Active learning (Line 369).*
The sentence “This robustness highlights the practical utility of our approach” is too vague.
Which specific approach or method is being referred to? Please clarify this explicitly.

*6. Definition of confidence scores (Line 430).*
The term “confidence scores” is used without a precise definition.
Please specify how these scores are computed, and how they relate to the rest of the methodology.

*7.* How is your method proving to be more efficient? By maintaining accuracy and decreasing ambiguity? It should be explicitly said.

---

> ### Author Response · Authors · 2025-11-22
>
> We thank the reviewer for their feedback on presentation and clarity, and address them below. Our discussions here have also been incorporated to create the revised version of the paper now updated on Openreview, with revised parts marked in color blue.
>
> > How is your method more efficient?
>
> Our method enables practitioners to anticipate downstream multiplicity without the need to explicitly create the Rashomon set. This avoids the prohibitive computational cost and practical infeasibility of training the large number of models required to approximate a Rashomon set for every single data processing decision. Instead, our framework provides a principled way to assess how data-processing decisions influence multiplicity upfront.
>
> Moreover, we introduce multiplicity-aware strategies that match the computational cost of standard baselines while leveraging our framework to intentionally steer the process toward higher or lower multiplicity. We have now added this clarification when defining the ‘Objective’ (Lines 210-215), as well as in the Conclusions section of the revised paper (Lines 516-520).
>
> > Clearer distinction between model parameters and the prediction functions.
>
> We have revised the notation. All instances that previously referred to the prediction function as θ() have now been replaced with f_θ() (Lines 187 and 685).
>
> > Leverage point in linear regression
>
> It is unclear to us how leverage points are connected to our work, and we would appreciate more clarification.
>
> > Relationship to stability-based approaches
>
> If we understand correctly, the reviewer here is referring to works on the stability of ML training. This is exactly what the literature on multiplicity studies, as discussed in our Related Works. We have further added explicit mention of stability-based approaches in the Related Work section of the revised version of the paper (Lines 128-129).
>
> > The notation 0^i_train is unclear. The definition of OVL^1_train is missing.
>
> We define 0^i_train and 1^i_train together in Lines 241-242, as the subset of the data with labels 0 and 1 respectively. We have now separated the definitions in the revised version.
>
> We define OVL^i_train in Line 252. The definition has been updated based on the comments about probability density estimates by another reviewer.
>
> > Which probability density is used?
>
> For our theoretical analysis, we assume true underlying probability distributions. For our empirical study, we estimate the probability density functions using histogram-based binning, which is a standard and well-established approach. We have now added these details explicitly in the revised version (Definition 4.1, Section 5.2 Experiment Setup, and Appendix C.1).
>
> > Correlations are stronger for LR and MLP
>
> We don’t believe that the correlation coefficient between downstream multiplicity and overlapping coefficients being better at capturing linear trends is related to linear dependencies captured by LR and MLP (one is the relationship between multiplicity across multiple models and overlapping coefficient of the data, and the other is the inductive bias of a single model). We would appreciate more details if the reviewer strongly believes in this explanation.
>
> However, our existing explanation was also only one potential possibility as mentioned in the paper. To avoid any confusion, we have now removed our explanation from the revised version of the paper, and we focus only on showing the results (Lines 410-411).
>
> > “This robustness highlights the practical utility of our approach” is too vague.
>
> As we discuss in Section 4.2, while our theoretical proofs provide an intuition for the trends of multiplicity, the assumptions of these proofs are quite strict, and thus not directly applicable in the real world. To support our hypothesis that this relationship between overlapping coefficient and multiplicity would still apply in the real world, we provide extensive empirical results in Sections 5 and 6. We find through two separate applications, that the empirical trends match our theoretical expectations.
>
> In lines 416-418, where we mention that “This robustness highlights the practical utility of our approach”, we are talking about our multiplicity-aware techniques MultLow and MultHigh, i.e., ‘our approach’ of anticipating multiplicity based on overlapping coefficient trends. We argue that since the trends in our experiments match our theoretical proofs, even though these settings do not follow the strict theoretical assumptions of our analysis, this highlights that our expectations of the relationship between overlap and multiplicity are ‘robust’ and can provide ‘practical utility’ for developers as a reliable measure of downstream multiplicity.
>
> > Confidence scores
>
> Confidence score refers to the model’s prediction probability, now clarified in the revised version of the paper (Lines 373-374).

---

### Official Review · Reviewer_nYro · 2025-10-31

**Soundness:** 3
**Presentation:** 3
**Contribution:** 2
**Rating:** 2
**Confidence:** 5

**Summary:**

The paper studies the predictive multiplicity (the extent to which training data allows many different high-accuracy prediction models) through the lens of neighboring datasets. It establishes a theoretical connection between the Rashomon parameter and the overlapping coefficient, so as to provide more insights on how the high overlapping of a dataset can lead to a small ambiguity parameter and thus is more beneficial to the learning problem.

**Strengths:**

The problem is well-motivated, and the paper is well-written and easy to follow.

The notion of neighboring dataset connects many different aspects: Rashomon parameter, ambiguity, and active learning.

**Weaknesses:**

My main concern is on the overlapping coefficient.

- The key parameter of the whole paper is this overlapping coefficient defined in 4.1.
- However, the useful quantity is defined on the training data, i.e., $OVL_{train}^i$. As I understand, for most datasets, this quantity is simply zero. To see it, the quantity is defined upon the empirical training distribution. Thus this quantity is only nonzero when the training dataset contains >= 2 samples with same x but different y, say Sample 1 (x, y=0) and Sample 2 (x, y=1) where both samples have the same feature. In practice, this rarely happens for (i) many of the features are continuous-valued, and (ii) there are a lot of features. In this case, the result provides no insights.

As a result,
- the framework is hardly generalizable to the regression setting where y is continuous; and it'd be possible but awkward to be generalized to the multiclass classification setting where y is multi-category.

For the numerical experiments,
- I guess the above factor is the reason for the choice of the three datasets but can't be applied for more general datasets such as UCI-repo or other deep learning datasets.

**Questions:**

See above. I spend quite amount of time reading the paper but might have overlook certain aspects; look forward to seeing the authors' response.

---

> ### Author Response · Authors · 2025-11-22
>
> We thank the reviewer for their time, and we are glad they found our work well-motivated and well-written. Below, we address the concerns raised by the reviewer. Our discussions here have also been incorporated to create the revised version of the paper now updated on Openreview, with revised parts marked in color blue.
>
> > Overlapping coefficient
>
> We appreciate the opportunity to clarify the calculation of the overlapping coefficient (OVL). As we define in Definition 4.1 (taken from (Inman & Bradley Jr, 1989)), OVL is calculated between two ‘probability (density) distributions’, not the pointwise distribution of the data.
>
> Indeed, the reviewer is correct that if the calculation was done on empirical training data with pointwise probabilities, the probability mass of overlap at any point would be zero unless two points overlap exactly. However, this is not how OVL is calculated. To calculate it, we first convert the empirical data into density estimations, using the standard approach of histogram-based binning, and then measure the overlap between the two density distributions. Thus, OVL provides insights into the overlap between the two distributions in binary classification, i.e., the ‘difficulty’ of the dataset.
>
> We have now added these details explicitly in the revised version (Definition 4.1, Section 5.2 Experiment Setup, and Appendix C.1).
>
> > Generalizability to regression setting and multiclass classification setting
>
> We use OVL to measure the ‘difficulty’ of a dataset. This has been previously measured in multiplicity literature as ‘noise in the data’ by Semenova et al. (2024) and ‘separability’ by Watson-Daniels et al. (2023b). Under binary classification, these terms are closely related (see several of them listed here - https://en.wikipedia.org/wiki/Total_variation_distance_of_probability_measures), including overlapping coefficient (which is simply 1 - total variation distance), and hence allows a more consistent evaluation of the ‘difficulty’ of a dataset.
>
> As we move to other settings, the measure of the ‘difficulty’ of a dataset gets complicated, and there isn't consensus in the literature for the same. However, the intuitions behind our framework still remain: if data processing choices increase the ‘difficulty’ of a dataset, the error for all models is likely to increase, creating a smaller Rashomon set under a fixed threshold. Thus, we believe our framework can be generalized to other settings in future under appropriate definitions of dataset difficulty.
>
> Finally, we would like to emphasize that our focus on binary classification is not a weakness, but a choice to provide deeper analysis. Many real world applications are binary classification, even more so in highly critical domains like medical diagnosis, loan applications, resume screening, etc., and thus our work can have significant real world impact. Moreover, both works most closely related to ours, by Semenova et al. (2024) and Watson-Daniels et al. (2023b), also focus specifically on binary classification, for similar reasons as above.
>
> To sum up, our framework is generalizable, although this is not trivial and requires an appropriate definition of ‘dataset difficulty’, which is itself an open problem (Ethayarajh et al., 2022). We focus specifically on binary classification, which allows us to provide a deeper analysis into the multiplicity trends. We have now added these discussions in the Conclusion section of the revised version of our paper (Lines 533-539).
>
> > More general datasets.
>
> We believe our experiments are quite robust, done not only on three datasets, but also across various applications (active learning and data imputation), random seeds (i.e., random starts in active learning and random missing values in data imputation), hyperparameters, and models. Consistent trends across these settings provide strong evidence for our hypothesis.
>
> Our experiments are, however, focused on tabular data and binary classification. While we don’t expect a change in modality to alter the trends, to further support the robustness of our study, we are extending our results to other modalities in the rebuttal phase. Within the next few days, we plan to provide additional results for a binary classification subset of the CIFAR10 dataset. Running all sets of experiments is expensive, since we need to create the Rashomon sets to show that our estimation of multiplicity is reliable. However, we will provide a small set of preliminary results during the rebuttal, and we will continue to run these experiments for the final version of the paper.
>
> As for extension beyond binary classification, as we discussed above, it is possible to generalize our framework in the future, but these extensions are beyond the scope of this paper.

---

> ### Author Response · Authors · 2025-11-27
>
> As promised, we provide additional preliminary results on the CIFAR-10 dataset. We take a binary subset of the dataset, focusing on two classes: dogs and cats. This is a common choice in the literature when working with a binary subset of CIFAR-10, as these two classes are shown to be the most difficult label pairs to differentiate in CIFAR-10 (Zhang et al., 2018). We use VGG16 as our model, trained from scratch on CIFAR-10 (i.e., without any pretrained weights). We use training configurations from the following open-source repository for our setup: https://github.com/kuangliu/pytorch-cifar
>
> We perform active learning with a similar setup as in our paper. We sample D0 of size n={500, 1000, 2000}, and the rest of the dataset is used as an unlabeled pool. We run active learning for up to 5 steps with query size q=100 per step. The complete pipeline is repeated for 10 random seeds. Directly measuring overlapping coefficient between high dimensional inputs is impractical, and instead, we use FID (Fréchet Inception Distance by Heusel et al., 2017) to measure the similarity of the two classes, i.e., the ‘difficulty’ of the dataset. Below, we provide the correlation scores between negative of FID (as it is a distance, not the overlap) and resulting multiplicity during active learning across various algorithms for the CIFAR-10 binary subset (corresponding to Figure 3 results present in our paper).
>
> |   | 1     | 2     | 3     | 4     | 5     |
> |------|-------|-------|-------|-------|-------|
> | 500  | -0.72 | -0.68 | -0.47 | -0.49 | -0.22 |
> | 1000 | -0.81 | -0.78 | -0.55 | -0.36 | -0.29 |
> | 2000 | -0.85 | -0.81 | -0.67 | -0.42 | -0.29 |
>
> These results support our original trends. We were not expecting any differences due to a change in modality, and thus, the results are in line with our hypothesis. Moreover, these results, although preliminary, also provide a glimpse into how our analysis can be generalized to new settings under appropriate definitions of 'dataset difficulty' (we use FID here instead of overlapping coefficients, which cannot be measured for high-dimensional datasets like images).
>
> References:
>
> [1] Zhang, Lingfeng, and Ioannis A. Kakadiaris. "A Hierarchical Matcher using Local Classifier Chains." arXiv preprint arXiv:1805.02339 (2018).
>
> [2] Heusel, Martin, et al. "Gans trained by a two time-scale update rule converge to a local nash equilibrium." Advances in neural information processing systems 30 (2017).

---

### Author Response · Authors · 2025-12-03
**Summarizing the Rebuttal Responses**

We thank all the reviewers for their feedback and time. Due to the unfortunate pause in any author-reviewer discussion, we take this opportunity to summarize our rebuttal efforts in this comment.

We have addressed all concerns raised by the reviewers in their respective comments and have revised the paper accordingly. During the rebuttal period, we were hoping to engage in further discussion, as we feel that the overall ratings given to our paper were harsher than the written comments, for instance, a score of 2 because of our focus on binary classification, or another score of 2 because of remarks on presentation. We believe our responses have adequately addressed the concerns raised, but given the lack of any possible revision of scores, we hope that the AC decision is not anchored to the original review scores.

**Reviewer nYro** found our problem well-motivated, our paper well-written, and agreed that the notion of neighboring dataset connects many different aspects: Rashomon parameter, ambiguity, and active learning. They had only one concern, the definition of the overlapping coefficient, and a followup concern that if the way they had understood the overlapping coefficient was correct, it would not be possible to generalize it to other settings. In our revision, we clarified this concern by showing that the overlapping coefficient is not calculated on the empirical pointwise distribution, but instead on the probability (density) distribution, using histogram-binning. We believe this also addresses their followup concerns, and we expand on this in further detail in our rebuttal. We even provided additional preliminary results that show both, (a) generalization beyond tabular datasets, and (b) generalization to settings where the overlapping coefficient cannot be calculated, and instead other approximations need to be used.

**Reviewer HugR** felt that our work addresses an interesting and practical problem. Their comments, while many, were all focused on the clarity of presentation. We have revised the paper to incorporate their suggestions, making the definitions more rigorous and using consistent notations. The comment that could have benefited from further discussion was on related work, where (a) we did not understand why the reviewer felt our work is related to the leverage point in regression, and (b) while we addressed their comment on stability-based approaches, we would have appreciated a confirmation that we had understood them correctly. Unfortunately, this cannot happen anymore, but we believe we have addressed all concerns raised by the reviewer to the best of our understanding.

**Reviewer xCmH** found our paper particularly clear, despite the number of notions introduced, and they also appreciated the explicit discussion of assumptions. They found our related work thorough and our findings interesting, given how it reversed expected trends. Their main concern was whether we should even consider maximizing or minimizing multiplicity. As we discuss in our rebuttal, and have added it to the revised version of our paper, the debate on whether to maximize or minimize multiplicity is one that is currently ongoing, and we don’t aim to take any sides here. Instead, we only provide a tool that developers can use to understand the impact of their choices on downstream multiplicity, a tool we believe would be valuable irrespective of which side of this debate one lands. Finally, the reviewer also had questions about the intuition behind why a smaller Rashomon set would likely lead to less multiplicity, and about the multiplicity scores of MultHigh, both of which were addressed during the rebuttal.

**Contributions:** We would also like to take this opportunity to re-emphasize the contribution of our work. Despite its limited scope, our work is the first in the literature to provide a framework for studying the impact of data processing choices on downstream multiplicity, which we believe is valuable and foundational for future work. The field of multiplicity has grown rapidly, yet as we discuss in our Related Works, there are very few papers that attempt to understand the relationship between data and multiplicity, and none that provide a framework that can be used by developers to understand the impact of their data processing choices on multiplicity. Not only do we provide theoretical analysis to establish clear trends, we also show the robustness of these trends in real-world datasets and across two separate data processing applications: active learning and data imputation. We outline several promising future research directions in our Conclusions section, which has been expanded further in the revised version based on the feedback provided by the reviewers. Overall, we believe our work here presents a strong first step towards understanding how data processing decisions impact multiplicity, without the prohibitively costly step of explicitly creating the Rashomon set.

---

### Meta-Review · Area_Chair_gdBo · 2026-01-04

**Summary:**

The paper investigates data-preprocessing-induced multiplicity by establishing a theoretical connection between the Rashomon parameter and the overlapping coefficient. Further, this framework is leveraged to active learning and data imputation, with an analysis of multiplicity in these areas and introducing multiplicity-aware methods for both tasks.

However, several issues remained after the rebuttal:
- Overlapping coefficient defined in 4.1. Some reviewers raises doubt on the validness of setting this parameter;
- Multiplicity-aware data acquisition and imputation strategies. It seems that authors aim to response this question, while their evidence is another reference, without any experimental or theoretical support.

Therefore, I recommend the reject, and I think that more clarification on the overlapping coefficient, with stronger motivation of multiplicity-aware data acquisition can improve this paper.

**Reviewer Concerns:**

- Overlapping coefficient defined in 4.1. Some reviewers raises doubt on the validness of setting this parameter;
- Multiplicity-aware data acquisition and imputation strategies. It seems that authors aim to response this question, while their evidence is another reference, without any experimental or theoretical support.

**Reviewer Scores:**

No reviewers will change their score given the concurrent rebuttal.

---

### Decision · Program_Chairs · 2026-01-26

Reject